# Community Empowerment Assessment and Community Nursing Diagnosis for Climate Change Mitigation and Adaptation in the Northern Region of the Portuguese Atlantic Coast: A Mixed-Methods Study Using MAIEC Framework

**Maria João Salvador Costa *** and **Pedro Melo**

Centre for Interdisciplinary Research in Health, Institute of Health Sciences, Universidade Católica Portuguesa, Rua de Diogo Botelho 1327, 4169-005 Porto, Portugal; pmelo@ucp.pt
* Correspondence: mjvcosta@ucp.pt

**Abstract:** The Community Intervention and Empowerment Assessment Model (MAIEC) offers a framework for community empowerment in several fields such as Climate Change (CC), the largest health emergency crisis globally, through diagnosis and interventions in Community Health Nursing. This study aims to assess the level of community empowerment in climate change mitigation and adaptation, and to identify nursing diagnosis through the MAIEC clinical decision matrix, within a local intermunicipal association in the northern region of the Portuguese Atlantic Coast. A convergent mixed-methods design was used, applying a focus group technique to a purposive sampling of ten key stakeholders of this community. A Portuguese version of the Empowerment Assessment Rating Scale and a questionnaire were both applied to the same participants, and qualitative and quantitative data generated were analysed using a content analysis technique and an Excel database sheet created using Microsoft Office 365. The analysis of the Portuguese northern community exposed: a low level of community empowerment for mitigation and adaptation to climate change; a nursing diagnosis of community management impairments in several dimensions, such as community process, community participation and community leadership. However, the study confirmed that MAIEC contributed to future community-based solutions, responding to the challenges of climate change, and enabling the planning of interventions to address MAIEC diagnoses in the form of CC-specific training and recommendations for new cooperation approaches from all stakeholders. This study was not registered.

**Keywords:** community empowerment status; community participation; stakeholders; climate change mitigation; climate change adaptation

## 1. Introduction

The present study is integrated with the Project HAC4CG_6—Heritage, Art, Creation for Climate Change. Living the city: catalysing spaces for learning, creation, and action towards climate change RL3 WP 6 is financed by the European Fund for Regional Development (FEDER) through the Regional Operational Program of the North of Portugal. It is aligned with the mission of Horizon Europe 2021–2027 and with the EU Green Deal. Its main aim is to identify the role of local governance, institutions and communities on climate change mitigation and adaptation.

In Portugal, some municipalities associate themselves to allow the coordination of actions among their geographical areas, to allow a common voice to their region, promoting negotiation and consensus for the execution of local policies, as well as the management of European funds, according to their specific needs. For a long time now, the European Union (EU) has been advocating a place-based approach concerning community funds management, and at present, associations of local municipalities such as the one targeted

by the current study—which will not be named for confidentiality purposes—have been directly designated by the government, and have the administrative and financial autonomy to serve their communities as appropriate [1].

To approach the community as a client, Nursing theoretical models can be applied, as some approach the community as a "*care unit*", using community empowerment strategies to tackle several challenges such as climate change. An example of such models is the *Community Assessment, Intervention and Empowerment Model* (MAIEC) [2], which helps Community and Public Health Nurses engage with their clinical interventions and supports their decision-making processes (please see Figure 1 below):

**Figure 1.** The Community Assessment, Intervention, and Empowerment Model (MAIEC)'s nursing diagnosis matrix for a community [3].

Within this Model, "Community Management" is the central diagnostic term, based on the International Classification for Nursing Practice (ICNP) [4]. It has three different dimensions:

1. Community process.
2. Community participation.
3. Community leadership.

MAIEC, as a theoretical Nursing Model, frames its paradigmatic concepts in Nursing by defining Community, Community Health, Community Environment, and Community Health Nursing Care as its main concepts [2]. As a Community Empowerment theoretical Model, it is based on the Community Empowerment concept which has been studied since the 1980s, first by Rappaport; then by Checkoway, Shulz, and Zimmerman in the 1990s; and more recently by Laverack [2]. Since earlier, researchers have supported a shared vision in which this concept corresponds to the cohesion of the members of the community, as well as its capacity to identify and autonomously manage their own issues [5].

Aligned with the missions of Horizon Europe [6] and with the Green Deal [7], the target chosen for our research is a community that includes three urban municipalities of the Portuguese coast in the northern region of Portugal. These three municipalities are deeply affected by unique climate characteristics induced by the adjacent coastal areas. Hence, the project *HAC4CG_6—Heritage, Art, Creation for Climate Change. Living the city: catalysing spaces for learning, creation and action towards climate change RL3 WP 6*, based at Universidade Católica Portuguesa, has led to the present study, offering an opportunity to understand the role of the local governance, institutions and communities in the mitigation and adaptation to climate change. The objectives defined for this project were:

1.  To diagnose and identify current practices and benchmarking with the best international practices (this was already carried out through a literature review developed and published in 2022);
2.  To develop political and ethical implications and recommendations to promote the role of local governance and institutions in climate change mitigation and adaptation (to be carried out by developing a piece of work that emphasizes best practices, which can be shared with the institutions involved);
3.  To develop Community Empowerment processes (the present study reflects the work already carried out in the three communities mentioned above, where a set of focus groups were implemented to study the community empowerment status, hopefully leading to commendations for the future of these communities).

Although the concentration of greenhouse gases (GHGs) in the atmosphere has risen since the 1850s, it is understood that this is mostly as a result of human activity [8]. Chemicals, fossil fuels, the industry, and the inappropriate use of land in agriculture, as well as deforestation, resulted in global warming and the consequent climate changes. The rise in global temperature as well as in the sea level has had great consequences both for the planet and for humans and their health [9].

According to a recent study published in 2022 [10], it is understood that urban areas on the Portuguese Atlantic Costal area have been greatly affected by, amongst other things, air pollution and its impact on climate. Reports of five thousand deaths have been associated with Nitrous Oxide ($NO_2$), small particles (PM2.5), as well as increased GHG emissions from urban transports and industry, ozone ($O_3$) included.

Therefore, there seems to be a window of opportunity in these coastal communities and contexts to enable specific interventions from within the various sectors of the community towards tackling climate change, either via mitigation or adaptation strategies.

Although top-down (government-based) interventions have been mostly used since 2008 (such as deep decarbonization plans in cities), Community Intervention Management Models of empowerment and resilience are now enhancing these processes by recommending bottom-up—local participation—interventions (such as local institutions, which usually are more focused on the GHG impact on the population), acknowledging qualitative data as highly significant by involving both experts and nonexperts who can easily recognise local meanings of resilience and circumstances in each community [11]. This is also supported by the MAIEC Model, in which the relevance of hybrid approaches is mentioned as part of a solution for early interventions on urban governance and climate change [2].

As a result, we can deduce that governmental institutions are not the only ones necessary for these processes. Nongovernmental organisations, the civil society, academia, the public health sector, as well as the private segment of society have also been acknowledged as significant stakeholders in climate change mitigation and adaptation [12]. Hence, the researchers of the present study have gathered several elements of the identified key sectors of society, named as key stakeholders, by setting up two focus groups in order to respond to the aims of this study:

1.  To assess the level of community empowerment regarding climate change mitigation and adaptation.
2.  To identify nursing diagnosis related to the community management foci in the context of climate change mitigation and adaptation by applying the MAIEC clinical decision-making matrix among the key stakeholders of the targeted community, which is located in the northern region of the Portuguese Atlantic Coast.

Recalling that urban areas are responsible for around 71–76% of global emissions, as the largest place-based source of GHG emissions [13], this study is clearly a priority, as it acknowledges the urban community empowerment status of this Portuguese Atlantic Coast region in regard to its community mitigation or adaptation actions.

The researchers hope that this study shows the importance of bringing the community together and of community empowerment, following Laverack's considerations on how

community empowerment actions can be a good strategy to promote health literacy within the community [14]. This is key to climate mitigation and adaptation strategies.

## 2. Material and Methods

Wishing to respond to the 13th Sustainable Development Goal (SDG) [15], the present study assumed that a positivist paradigm perspective should be complemented with a constructivist paradigm [16] to develop knowledge and insight based on the reality of an urban community. Instead of being limited to conventional research, where research can be conducted solely aiming at an individual lifestyle approach, adding a dialectic methodology and studying the socially constructed realities of this coastal community can enhance valuable local outcomes and gains in health and illness prevention. Both ontological, epistemological, and methodological underpinnings can be used together to achieve a broader understanding of the local "level of empowerment". Although the use of both paradigms seems to be a way forward, as it allows researchers and individuals to interact with themselves and with their own community or environment whilst using the research process, their complementarity cannot always set a universal framework for all communities, as it is based in the singularity of this specific community.

Whilst participating in creating this particular group of key players—the stakeholders—the researchers proceeded with a mixed-methods study using a convergent parallel design to be carried out within these three urban municipalities of the northern region of the Portuguese Atlantic Coast. The study followed the good reporting guidelines of the Application of Mixed Methods in Health Services Management Research study and ran from September 2022 to March 2023, having been approved by the Ethics Committee in Technology, Social Sciences and Humanities of Universidade Católica Portuguesa.

According to Creswell's typology [17], basic mixed-methods research, using a convergent parallel design, combines both quantitative and qualitative methods to broaden the understanding of a phenomenon within a chosen population, enabling two different perspectives [18]. Please see Figure 2 below, which illustrates the relationship and sequence of both research components:

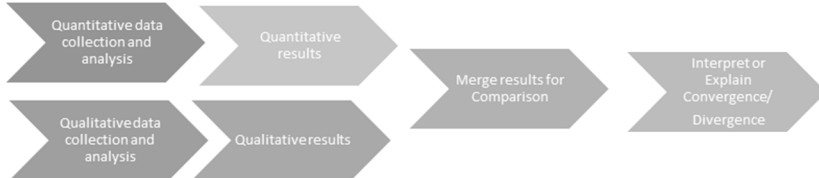

**Figure 2.** Convergent parallel design [19].

A purposive sampling of ten participants from the chosen intermunicipal association, representing each sector of the society, previously identified as key stakeholders within the most current literature reviews, has taken part in the two focus groups organised, following the inclusion criteria below:

- The participant works in institutions based in one of the three municipalities targeted by the study.
- The participant is thoroughly informed and consents to enrol in the study.
- The participant has been identified as a key stakeholder with regard to climate change mitigation or adaptation in this community, as per the literature review.

The process of organizing both focus groups started with the determination of the type and number of participants, the time length of the sessions, as well as how these would be structured.

The recruitment of the 10 stakeholders was carried out via email invitation, following thorough research of the community's key stakeholders per area (local leaders, local government and nongovernment representatives, local institutional managers, academics,

health professionals, and others), who led on or implemented community actions towards climate change mitigation and adaptation.

Additionally, a questionnaire developed with an online survey creator—Microsoft Forms, part of Office 365—was also applied to all participants ($n = 10$), with the purpose of searching for quantitative and statistical patterns and gaps related to participants' experience, knowledge, and beliefs on Climate Change mitigation and adaptation, where qualitative open-ended questions were also added to the questionnaire. Data analysis was then conducted accordingly, using a content analysis methodology for the qualitative part of the study and an Excel database sheet from Microsoft Office 365 version to format, organize, and calculate data from the quantitative part of the study. Both data collection and analysis were carried out by one of our research fellows, a student enrolled in a PhD in Nursing at Universidade Católica Portuguesa, namely a Community and Public Health Nurse specialist.

The focus group interviews were conducted online using Zoom. First, all ten participants answered questions from a LIVE discussion script—the Portuguese version of the *Empowerment and Assessment Rating Scale* [2], adapted to the climate change subject. Second, they responded to a questionnaire related to each one of the diagnostic dimensions of community management, assessing trends on *Community Leadership* (knowledge, beliefs and behaviours associated with climate change mitigation and adaptation interventions), *Community Participation* (perception of the existence of organisational structures and partnerships to promote climate change mitigation and adaptation), and *Community Process* (previous experiences with climate action). The questionnaire included single choice, multiple choice, and open questions, as well as rating scales.

*Ethics*

To prevent scientific misconduct and secure research fundamental ethical principles, both instruments were applied while ensuring that confidentiality and anonymity of all participants' answers were maintained at all times. Although every participant of each focus group answered in an ONLINE LIVE Zoom meeting, all responses have been anonymously downloaded as a report to simplify data collection.

All participants have been previously and fully informed that anonymity would be assured, why the research was being conducted, and how their data would be used, considering that there were no risks associated with their participation within the study.

Both sessions were structured and organised as per Phase 1 and 2, below, where both research surveys were applied:

- Phase 1—To assess the empowerment status of the community regarding climate change mitigation and adaptation, the researchers proceeded with the application of the Portuguese version of the *Empowerment and Assessment Rating Scale*, which was already validated. This instrument contained nine domains, each designed to capture different information related to the level of empowerment.
- Phase 2—The researchers applied a questionnaire using the MAIEC clinical decision-making matrix to identify nursing diagnoses related to community management, allowing the assessment of each one of the diagnostic dimensions of community management.

To assess the level of empowerment of these communities, and to identify the diagnoses of community management, a combination of both quantitative and qualitative methods was required, with the purpose of collecting data on the participants' points of view and experiences.

As part of the MAIEC protocol, the Portuguese version of the Community Empowerment Rating Scale [2], was applied to all participants ($n = 10$) as a discussion script used by a focus group. Data collection was therefore performed using this qualitative interview technique where informal discussion took place.

### 3. Results

Although the research team attempted to gather a larger sample of participants in each group, in total, only ten key stakeholders on climate change mitigation and adaptation accepted the invitation to take part in the present study. Regardless, the ten participants recruited to join the two focus groups were carefully designated, ensuring that both genders and a variety of relevant stakeholders were appropriately and reasonably represented. Of the ten elements interviewed, half ($n = 5$) were female.

These participants represented key institutions or organizations that play a crucial role in tackling climate change within the communities targeted by our study. Amongst the two groups of participants, 30% ($n = 3$) were representatives of the local municipalities (one architect and two engineers); 30% ($n = 3$) worked as specialist nurses in local Public Health Units; 20% ($n = 2$) represented Emergency/Civil Protection Operational Services; 10% ($n = 1$) were representing the Academic Sector (one senior university lecturer); and 10% ($n = 1$) were local leaders specifically representing the Nongovernmental Organisations (NGOs) within the targeted municipalities. Please refer to Table 1 below, which displays the types and number of stakeholders represented amongst all participants involved in both focus groups:

**Table 1.** Representation of key stakeholders amongst the participants interviewed for the focus groups.

| Stakeholders | *n* |
|---|---|
| Public Health Sector | $n = 3$ |
| Local Municipalities | $n = 3$ |
| Emergency and Civil Protection Services | $n = 2$ |
| Universities | $n = 1$ |
| Nongovernmental Organisations (NGOs) | $n = 1$ |

Following the application of the Portuguese version of the *Empowerment and Assessment Rating Scale (EavEc)* [2] amongst the ten participants interviewed in the focus groups, the study has shown that the *level of community empowerment* for climate change mitigation and adaptation within these three municipalities of a northern region of the Portuguese Atlantic Coast is categorised at the lowest level (level 1) of the above-mentioned scale.

Please refer to Table 2 below, which demonstrates how domains such as community participation, problem assessment capability, local leadership, organisational structures, resource mobilisation, external linkages, critical awareness, project management, and relation with external agents have been assessed and categorised.

**Table 2.** Level of community empowerment for climate change mitigation and adaptation within the intermunicipal association in a northern region of the Portuguese Atlantic coast, following application of the Portuguese version of the Empowerment Assessment Rating Scale.

| EAvEc Domains | Level of Community Empowerment for Climate Change Mitigation and Adaptation *. Rating Scale (1—Low to 5—High) |
|---|---|
| Community Participation | 1 |
| Problem Assessment Capability | 1 |
| Local Leadership | 1 |
| Organisational structures | 1 |
| Resource mobilisation | 1 |
| External linkages | 1 |
| Critical Awareness | 1 |
| Project Management | 1 |
| Relation with external agents | 1 |

* According to the Portuguese version of the *Empowerment and Assessment Rating Scale* [2].

According to Table 2, the level of empowerment of these coastal communities is categorised as of Level 1, at a scale from 1 to 5, in the following domains:

1. *Organisational structures*—The participants of the study were unable to identify any organisational structures within the community, where all of them are represented, in a work group or committee.
2. *Resource mobilisation*—The participants stated that resources are not being mobilised within the community, considering the representatives attending both the focus groups.
3. *External Linkages*—There are no external links involving all the representatives within this community.
4. *Critical Awareness*—The participants feel that, as a community, there is no awareness on the local problems, and therefore there are no group discussions on the topic.
5. *Project/Program Management*—According to the answers of the participants, they have not yet identified any existing project/program involving all the stakeholders in this community represented in the focus groups, nor any external agent appointed to address the problems related to climate change.
6. *Relation with external agents*—The participants of the study denied the existence of any policies, finances, resources, or any program assessment developed by all the community actors representing the three municipalities, and using a common approach.
7. *Community Participation*—The participants representing each institution within the three municipalities are not currently participating in shared meetings related to CC mitigation and adaptation.
8. *Problem Assessment Capability*—There seems to be no problem assessment developed in cooperation by this community, up until the date of each focus group (December, 2022), according to the participants.
9. *Local Leadership*—At this point, the community does not have any structured organisation working in cooperation towards climate change mitigation and adaptation; therefore, the participants were not able to identify any designated leader thus far.

Additionally, and according to the data collected for the assessment of the community management focus, the present study has identified that *Community Management* is **impaired** within the three municipalities targeted by the study.

Likewise, the three diagnostic dimensions proposed by MAIEC are also **impaired**, as thoroughly described and summarized in Table 4:

1. The ***Community Process*** is **impaired**: only twenty-five percent of the participants have *previous experiences* with similar projects, meaning that most of the elements within the two sessions have no connections whatsoever with other stakeholders and agents of the community to approach climate change issues.
2. The ***Community Participation*** is **impaired**, based on the following assumptions:
   a. *Organisational structures*—40% percent of the participants could not identify any organisational structures aiming at climate change mitigation or adaptation; however, the remaining participants have identified six work groups/commissions as established structures within the three targeted municipalities, whose goal is to look out for solutions on climate change concerns. These were identified as the "Pacto do Porto para o clima", "MovRioDouro", "Centro de Investigação em Saúde e Ambiente, Escola Superior de Saúde do Instituto Politécnico do Porto", "Estratégia Municipal de Adaptação às Alterações Climáticas", "Cidamb", and "Grupo das Alterações Climáticas da ANMP".
   b. *Communication*—The community and its leaders seem to have ineffective means of communication, as only 40% of the participants have acknowledged familiarity with the existence of means to clearly communicate climate change issues with leaders or other agents within the organisational structures; 20% are not aware of any specific way of communicating these issues and the remaining 40% did not answer this question.

c.  *Partnerships*—Only half (*n* = 5) of the participants are fully aware of existing resources to form partnerships to solve any problems which may arise, having identified structures such as the Ministry of the Environment (Parque Esponja da Asprela), STCP and METRO, Cooperativa Coopernic, INEGI, INESCTEC, EFACEC Energia, Electric Mobile, VPS, EVIO Federação Académica do Porto, International Development Norway: Aprela + Sustentável, Pacto do Porto para o Clima, APA, Projeto Quinto Alçado, Projeto Cidades Sustentáveis, and MCR2030 to form partnerships with.

3.  The *Community Leadership* is also impaired as clarified below:

(a)  *Leaders and participants' knowledge* of the problem was not demonstrated— 10% (*n* = 1) of the participants were unable to demonstrate their knowledge regarding the main factors leading to CC. Although all of them (*n* = 10) knew at least three consequences of CC, only 30% (*n* = 3) stated that the energy sector is the one that most contributes to GHG emissions (please see Table 3 below). Only 40% (*n* = 4) were able to identify ozone as a GHG, apart from methane and nitrogen dioxide. Although all of them were able to identify vulnerable groups from the population, 30% (*n* = 3) were still unable to identify the most common diseases caused by this global concern. Only 50% (*n* = 5) were aware of the deadline date established for the Paris Agreement towards carbon neutrality. Although 60% (*n* = 6) were fully aware of the main causes of small particles (PM2.5) atmospheric events, 10% (*n* = 1) still did not know what these are, or what actions need to be taken to avoid being exposed to it; nevertheless, all of the participants (*n* = 10) acknowledged the contributions of green spaces in cities towards CC mitigation. However, 10% (*n* = 1) still do not relate "*walking and cycling*" as legitimate means of tackling climate change.

**Table 3.** Main sectors contributing to GHG emissions, according to focus group analysis of key stakeholders within the three municipalities studied.

| Main Sectors Contributing to GHG Emissions | *n* |
|---|---|
| Industrial Processes and Products | *n* = 6 |
| Energy | *n* = 3 |
| Agriculture | *n* = 1 |
| Waste management | *n* = 0 |

(b)  *Leaders and participants' beliefs* around the project and their competencies at approaching CC—Only 10% of leaders and participants feel they have full knowledge of CC, whereas the majority (*n* = 6) believe they still require training on the subject, considering themselves to hold average knowledge on the field. With regard to CC mitigation and adaptation strategies, 70% of leaders and participants consider themselves to have medium knowledge or below, in contrast with a minority (*n* = 1) that feel fully confident regarding their knowledge on the means to tackle CC. Only one-fifth of the leaders and participants of the present study believes each citizen may actually make a difference with regard to CC and only 20% of the two groups are optimistic concerning the impact of their institution on CC mitigation and adaptation strategies.

Main diagnosis and diagnostic dimensions of community management were therefore identified within the three communities of the northern region of the Portuguese Atlantic coast, please see Table 4 below.

**Table 4.** Main diagnosis and diagnostic dimensions of community management identified within the three communities of the northern region of the Portuguese Atlantic coast.

| Main Diagnosis | |
|---|---|
| *Community Management Impaired* | |
| **Diagnostic Dimensions Diagnoses** | **Sub Diagnoses** |
| *Impaired Community Leadership* | Cognitive Dimension Impaired<br>• Knowledge regarding main factors leading to CC not demonstrated in 10% of the participants.<br>• Knowledge regarding the sectors most involved in GHG emissions not demonstrated in 40%.<br>• Knowledge regarding ozone as a GHG not demonstrated in 30%.<br>• Knowledge regarding the most common diseases caused by CC not demonstrated in 30%.<br>• Knowledge regarding the Paris Agreement deadline proposal for carbon neutrality not demonstrated in 50% of the participants.<br>• Knowledge regarding the several origins of small particles (PM2.5) not demonstrated in 40%.<br>• Knowledge regarding the contributes of walking and cycling to mitigate and adapt to CC not demonstrated in 10%.<br><br>Beliefs Impaired<br>• Participants' beliefs regarding their own knowledge in CC issues impaired in 60%.<br>• Participants' beliefs regarding CC mitigation and adaptation strategies impaired in 70%.<br>• Participants' beliefs regarding citizens' contributions towards CC mitigation and adaptation impaired in 80%.<br>Participants' beliefs regarding the impact of their institution on CC mitigation and adaptation strategies impaired in 80%. |
| *Impaired Community Participation* | <u>Ineffective Communication</u> demonstrated by 60% of the participants, who are either not aware of any means of communicating with CC external agents or did not reply.<br><u>Inexistence of Partnerships</u> in the perception of half of the participants of the study, who are unaware of existing resources to form any partnerships.<br><u>Inexistence of organisational structures</u> in the perception of 40% of the participants. |
| *Impaired Community Process* | Community Coping Impaired demonstrated by 75% of the participants, who have no previous experience with CC projects and work. |

## 4. Discussion

Firstly, analysing the data collected, the study demonstrates that the Level of Empowerment for climate change mitigation and adaptation within these three municipalities of a northern region of the Portuguese Atlantic Coast is effectively *low*.

Although there might exist secluded work conducive to climate change mitigation and adaptation, there are no synergetic efforts in place amongst the participants in terms of putting together the three municipalities, in which they work alongside each other as a group work or committee. It also seems that not all the members interviewed are participating in activities and meetings already set up by the community. They were not capable of assessing the current problems as a group. The participants were also

incapable of identifying any leader or leaders within their community, as no structured organisation has yet been created between them all. Although a few climate-related organisations have been identified within the community, they are working in isolation, without involving all key stakeholders in climate change mitigation and adaptation within the area. Resources available within the three municipalities seem not to have been yet jointly mobilised by its leaders and members in cooperation. Participants have not been able to link externally as a group with other climate agents or other organisations. Group meetings between the participants to discuss climate related issues within these three municipalities of the north of Portugal have not been yet set up. The management of climate-related projects and programs is not being performed by any organisation, nor any external agent appointed for this specific purpose; there is no evidence of relations with external agents in regard to decision making, and there seem to be no discussions amongst key players in the community concerning climate-related policies, finances, resources and project management.

Secondly, the *Community Management* for climate change mitigation and adaptation assessed within the participants of the two focus groups, which represented key stakeholders of the three municipalities targeted by the study, was classified as *impaired*, and unveiled an impairment of its three diagnostic dimensions: *Community Process, Community Participation and Community Leadership*:

(a)  The impairment of the Community Process, due to a lack of previous experience in the field, indicates that only a minority of these stakeholders have participated in other projects related to climate change.

(b)  With regard to Community Participation, it becomes evident that although most of the stakeholders could identify a few organisations with whom they were individually involved in tackling CC, nearly half were not integrated in any form of organisational structure or partnership, revealing a gap in the community concerning cooperative key strategies to promote the mitigation of and adaptation to CC. Additionally, a majority of the participants were unaware of means of communicating amongst the group, a key factor to discuss, plan and share synergies, when addressing climate change impact and risks.

(c)  As to the Community Leadership, and its cognitive dimension, the majority of the participants surveyed throughout the focus group seemed knowledgeable on the main factors contributing to CC, referring to human behaviour and human consumption, the use of nonrenewable energy, GHG emissions, air pollution and global warming, deforestation, transport, and fossil fuel combustion as the main causes for CC; however, nearly half of the participants are still not fully aware of the sectors with the highest level of involvement in GHG emissions, whilst evidence clearly shows that Energy and Industry and its processes and products are indeed the main causes of GHG emissions [20]. The majority of the participants in this study were also still unable to recognise ozone as a deadly GHG and pollutant, as clearly evidenced by contemporaneous research and data [10], apart from methane and nitrogen dioxide. The study also shows that within this group, a significant number of participants are, to date, still not aware of the most common diseases caused by CC, meaning that much is yet to be done concerning climate and health literacy amongst key stakeholders. Since the Paris Agreement, Europe has been mobilised by Brussels towards a greener continent, meaning that sustainability and carbon neutrality have become common concepts within organisations; however, this study demonstrates that regardless of the work initiated in Paris in 2015, some communities in European countries such as Portugal seem not to be fully informed on the shared target for 2050—only half of the group surveyed was familiar with the deadline date, 30% assumed this was 2030, and 20% were clearly unaware of any date whatsoever; with regard to the significant atmospheric events of PM2.5, known as small particles, which occur more frequently every year and have multiple causes, nearly half of the participants were still unable to identify its main causes: 20% seemed to exclusively associate PM2.5 with vehicle

and plane engines and 10% with wood waste burning. Thus, there is still a lot to be done in terms of empowering this coastal community, where 10% of the stakeholders surveyed were, at this point, still incapable of identifying the necessary actions to be taken to help the public avoid being exposed to these sort of events. Finally, a high point in the present study refers to the fact that all stakeholders seemed unanimous in regard to the benefits of using green spaces in the cities to mitigate the impact of CC; however, a minority remain unaware of "*walking and cycling*" as a mitigating strategy, as it promotes the reduction of circulation of cars, cutting down on GHG emissions, reducing noise pollution and congestion, whilst helping to reduce physical inactivity and sedentarism.

Several papers have indicated that MAIEC has previously been helpful in other contexts, showing several advantages of considering a community as a *nursing care-unit* [21]. Again, in the current project, the MAIEC theoretical model has shown to be helpful for identifying gaps within this particular coastal community concerning CC mitigation and adaptation, revealing that a community-based approach is required to effectively face the challenges that CC poses.

Nevertheless, a few limitations have been identified for the present study:

- The study merely refers to a specific sample of stakeholders, in a specific intermunicipal community, in a time-specific framework; hence, generalisations cannot be considered.
- Although some variables have been considered, such as a basic level of CC knowledge detained, others, such as academic qualifications, CC-related training received, etc., have been omitted from the study.
- Additionally, although a thorough explanation of the survey was carried out for the assessment of the community management, the researchers were unable to control personal and cultural characteristics that may have affected participants' understanding of each question.

Despite all the above-mentioned limitations, the authors of the present study agree that their work has, however, contributed to overseeing this coastal community as a *nursing care unit*, identifying nursing diagnoses that point out the need for clinical decisions that must implement new community-based interventions, ensuring citizens and stakeholders' participation and involvement, improving communication paths, organising structured partnerships, increasing the level of CC-base knowledge and training, preventing sceptical or inappropriate beliefs, and promoting alternative and climate friendly lifestyles and behaviours, which will ultimately and hopefully help individuals, families, populations, and communities tackle this global public health concern.

Another relevant point is the fact that the present research project was truly innovative, as it has never been undertaken in a similar way. Although climate change mitigation and adaptation have recently been embraced by nurses around the globe, the present work shows that a community-based nursing approach, performed by community and public health specialist nurses who apply the MAIEC decision matrix, enables the identification of relevant nursing diagnoses. This allows the detection of unknown gaps and windows of opportunity to implement new interventions essential to the enhancement of community management of this health emergency—this will ultimately reflect the community empowerment required to swiftly initiate new courses of action in mitigating and adapting to this global concern.

Still, for future reference and research, another assessment for the community management of climate change mitigation and adaptation should be carried out, using the same processes and amongst the same participants, to monitor the level of empowerment of this coastal community following the implementation of interventions proposed by MAIEC. For this purpose, the researchers fully recommend a follow-up study in order to properly reassess the impact of these interventions within the targeted community.

## 5. Conclusions

The data analysed in the present study revealed a *low level of empowerment for climate change mitigation and adaptation* within the targeted three municipalities in a northern region of the Portuguese Atlantic coast. It also identified a *nursing diagnosis of impaired community management* in the field, which reflected an impairment of all its diagnostic dimensions: community process, community participation, and community leadership.

Using an interdisciplinary approach and interviewing several key stakeholders, involving municipal architects, engineers, nurses, civil protection operational leaders, and academics was very significant to the research, and revealed a lack of experience, established partnerships, communication mechanisms, and knowledge amongst these professionals, in the discussion of climate change mitigation and adaptation.

Another key finding of the study is the impairment of the community's cognitive dimension, meaning that a majority of the participants of the focus group were knowledgeable on CC-related content, yet there are institutions and other community-based actors who still do not hold sufficient basic knowledge on the field to plan for adequate policies or interventions.

The results are consistent with other studies in the field in which it is acknowledged that global-level policy guidance is disconnected between primary healthcare and climate, recommending the empowerment of communities as well as the implementation of adequate multisectoral climate action [22]. These studies also support the need to urgently promote consensus for community-based interventions to allow new and urgent practices, involving all stakeholders in organised structures, amplifying synergies and resources available to tackle the problems caused by climate change.

Finally, we may add that the theoretical intervention model of MAIEC, as a key benchmark, enables the promotion of community-based solutions. For the future, the present study fully recommends specific CC-training and a new cooperation approach from all key stakeholders involved in climate change mitigation and adaptation within this local intermunicipal association in the northern region of the Portuguese Atlantic coast.

**Author Contributions:** Conceptualization, M.J.S.C. and P.M.; methodology, M.J.S.C. and P.M.; validation, P.M.; investigation, M.J.S.C.; resources, M.J.S.C. and P.M.; visualization, M.J.S.C. and P.M.; project administration, P.M.; funding acquisition, P.M.; writing—original draft preparation, M.J.S.C.; writing—review and editing, M.J.S.C. and P.M. All authors have read and agreed to the published version of the manuscript.

**Funding:** This research was funded by Fundação para a Ciência e Tecnologia (FCT), Portugal, Project HAC4CG_6-Heritage, Art, Creation for Climate Change. Living the city: catalysing spaces for learning, creation, and action towards climate change RL3 WP 6-HAC4-CG (NORTE-01-0145-FEDER-000067) and the APC was funded by one of the authors.

**Institutional Review Board Statement:** The study was conducted in accordance with the Declaration of Helsinki and approved by the Ethics Committee in Technology, Social Sciences and Humanities (CETCH) of Universidade Católica Portuguesa (protocol code: CETCH2023-43; date of approval: 20 April 2023).

**Informed Consent Statement:** Informed consent was obtained from all subjects involved in the study.

**Data Availability Statement:** Data from this study are available from the corresponding author upon reasonable request. The data are not publicly available due to privacy reasons.

**Public Involvement Statement:** No public involvement in any aspect of this research.

**Guidelines and Standards Statement:** This manuscript was drafted against the "Lee, S.D.; Iott, B.; Banaszak-Holl, J.; Shih, S.F.; Raj, M.; Johnson, K.E.; Kiessling, K.; Moore-Petinak, N. Application of Mixed Methods in Health Services Management Research: A Systematic Review. *Med. Care Res. Rev.* **2022**, *79*, 331–344" reporting guidelines.

**Acknowledgments:** The authors wish to thank all the support from the three municipalities that agreed to participate in the present study and be represented by their key stakeholders.

**Conflicts of Interest:** The authors declare no conflict of interest.

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
