# Peer review of "Community Empowerment Assessment and Community Nursing Diagnosis for Climate Change Mitigation and Adaptation in the Northern Region of the Portuguese Atlantic Coast: A Mixed-Methods Study Using MAIEC Framework"

_nursrep, doi:10.3390/nursrep13030085_

Round 1
Reviewer 1 Report
Dear Authors
Thank you for an exciting paper, however somewhat hard to read as it sometimes becomes too wordy. Which in turn hides the clarity of the message. I think the paper would benefit from rereading and clarifying some perspectives and wow these connect it to nursing care for the individual nurse. I would like the authors to develop on how the tpositivist paradigm perspective is complemented with a constructivist paradigm. How is this done, what are the theoretical underpinnings? the strengths and weaknesses? also would like to see a section about ethics.
Since this is a small material it would be informative to connect it to other studies in the same field.
Thank you
Reviewer 2 Report
In the abstract, you need some information about a sampling of the study.
in the secon aim of your study, you stated that " To identify nursing diagnosis related to the community management foci in the con- 125 text of climate change mitigation and adaptation, by applying ....._) if possible, please add something representative of this aim in the title because, in the title, nothing mentions about nursing
In the discussion or conclusion, you need to add something about the limitations of your study.
